# Nutritional Content, Phytochemical Profiling, and Physical Properties of Buckwheat (*Fagopyrum esculentum*) Seeds for Promotion of Dietary and Food Ingredient Biodiversity

**Madalina Neacsu** [1,*], **Shirley De Lima Sampaio** [1,2], **Helen E. Hayes** [1] 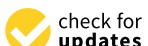, **Gary J. Duncan** [1], **Nicholas J. Vaughan** [1], **Wendy R. Russell** [1] **and Vassilios Raikos** [1,3]

1 Rowett Institute, University of Aberdeen, Foresterhill, Aberdeen AB25 2ZD, UK
2 Centro de Investigação de Montanha (CIMO), Instituto Politecnico de Bragança, Campus de Santa Apolonia, 5300-253 Bragança, Portugal
3 Department of Nutrition and Dietetics Sciences, School of Health Sciences, Hellenic Mediterranean University, 723 00 Crete, Greece
* Correspondence: m.neacsu@abdn.ac.uk; Tel.: +44-1224438760

**Abstract:** The adoption of food crops as a source of dietary macro- and micro-nutrients is a sustainable way to promote diet biodiversity and health while being respectful to the environment. The aim of this work was to comprehensively characterize the nutrient and phytochemical content of buckwheat seeds (*Fagopyrum esculentum*) and assess their physical properties for the evaluation of their suitability as food ingredients. The buckwheat samples were found to be complete sources of amino acids (UPLC-TUV analysis), with a protein content between $11.71 \pm 0.40\%$ and $14.13 \pm 0.50\%$ (Vario Max CN analysis), and a source of insoluble fiber with $11.05 \pm 0.25$ %, in the UK hulled samples (GC analysis). The buckwheat samples were rich in phosphorus, manganese, molybdenum, zinc, magnesium, and selenium (ICP-MS analysis). A total of 196 plant metabolites were detected using HPLC and LCMS analysis, with anthocyanidins (pelargonidin and cyanidin) being the most abundant phenolic molecules that were measured in all the buckwheat samples. Removing the hull was beneficial for increasing the powder bulk density, whereas the hulled buckwheat samples were more easily rehydrated. The implementation of buckwheat as a staple food crop has enormous potential for the food industry, human nutrition, and diet diversification and could contribute towards meeting the daily recommendation for dietary fiber, essential amino acids, and minerals in Western-style diet countries such as the UK.

**Keywords:** buckwheat; *Fagopyrum esculentum*; hulls; NSP; plant protein; amino acids; minerals; plant secondary metabolites; dietary biodiversity

## 1. Introduction

Increasing the consumption of plant-based foods is an effective method for tackling the global challenges of under- and over-nutrition and their associated impact on public health and environmental sustainability. Dependence on cereal-based diets in low- and middle-income countries, is considered a factor contributing to malnutrition, due to the relatively low levels of essential amino acids in wheat and maize [1]. Moreover, the Western-style diet, characterized by highly processed and refined foods, scarce in wholegrains and high in sugars, salt, and fat, as well as protein from red meat contributes to metabolic disturbances and the development of obesity-related diseases [2]. Diet macronutrient biodiversification (i.e., protein) to include plant-based sources could promote agricultural biodiversity and contribute towards meeting climate targets [3], as well as improving health [4].

Buckwheat (*Fagopyrum* spp.) is a pseudo-cereal belonging to the *Polygonaceae* family [5] and is cultivated worldwide mostly in Russia and China. The annual production of the pseudo cereal has risen from 2 million metric tons in 2014 to 3.8 million metric tons in

2017 [6,7]. In Europe, farming buckwheat as a food crop has been largely replaced by the production of cereals such as wheat [8]. In the UK, buckwheat is not farmed for food but for other uses: to enrich soil such as green manure, suppress weeds, and attract bees and other insect pollinators [8].

Buckwheat is a good source of protein with a desirable amino acid profile. Buckwheat proteins contain higher levels of sulfur-containing (cysteine and methionine) and essential (lysine and tryptophan) amino acids compared to rice or maize [9]. Moreover, it is rich in dietary fiber, lipids, minerals, and other bioactive components such as phenolic compounds and sterols [10–12]. Compounds such as flavonoids and phenolic acids are abundant in buckwheat, especially catechins, rutin, quercetin, caffeic acid, and syringic acid and anthocyanins are all reported in abundance in buckwheat [10,13]. These molecules are extensively studied for their health benefits such as antioxidative and hypoglycemic potentials. Previous research showed that buckwheat was very efficient at reducing hunger and promoting satiety, as well as delivering the lowest levels of circulatory branched chain amino acids when it was consumed by healthy volunteers compared with other plant and animal protein-rich meals [14].

Buckwheat seeds are either milled to a fine flour or dehulled and used whole (groat) for various culinary purposes. The culinary uses of the pyramid-shaped seed of buckwheat resemble typical cereal applications in foods and include bread, pasta, noodles, cookies, and pancakes [15]. Drying and milling the plant material to produce flour effectively preserves the nutritional profile and ease of transportation, storage, dosing, and mixing into food formulations leading to innovative formulations for a wide range of food and beverages [16]. However, dried foods must possess certain physical and sensory properties to be suitable for industrial or domestic use.

The aim of this work was to comprehensively characterize the nutrient and phytochemical content of buckwheat seeds (*Fagopyrum esculentum*) that were sourced from the United Kingdom (UK) and Brazil (BR); and assess their techno-functional properties for suitability as food ingredients. Assessing the nutritional and chemical profile of buckwheat and understanding its physical properties is essential for promoting dietary biodiversity and future farming potential of buckwheat as a food crop.

## 2. Materials and Methods

### 2.1. Materials

There were three buckwheat (*Fagopyrum esculentum, Polygonaceae*) samples that were sourced; one hulled and one dehulled from Protecta (Parana, Brazil) and one hulled sample that was supplied by The British Quinoa Company (Shropshire, UK). All the samples were freeze-milled (Spex sample prep 6800; Munich, Germany) and stored at room temperature in desiccators under vacuum. Standards and general laboratory reagents were purchased from Sigma-Aldrich (Gillingham, UK) and Fisher Scientific UK Ltd. (Loughborough, UK) or synthesized as described previously [17,18]. The chemicals that were used for ICP-MS analysis were nitric acid of TraceSelect Ultra grade (Fluka), hydrochloric acid (30%) of Ultrapur grade (Merck; Darmstadt, Germany), and deionized water (Millipore; Bedford, MA, USA). Single element standards were purchased from all Inorganic Ventures (Christiansburg, VA, USA).

### 2.2. Proximate Analysis

The protein was estimated as total nitrogen by the Dumas combustion method using a Vario Max CN analyzer (the nitrogen content was multiplied by 6.25 to estimate the protein concentration) [19]. Previously published methods [20] were used to determine the resistant starch and soluble and insoluble non-starch polysaccharide (NSP) content. The total fat content was determined by the Soxtec method (Soxtec™ 2050 Auto Fat Extraction System) [21].

### 2.3. Trace Element Analysis

Microelements analysis used a previously published method [10]. In brief: the samples (0.4 g, n = 3) were mixed with distilled water and nitric acid and heated following two temperature gradients, 20 °C–150 °C (15 min) and 150 °C–165 °C (10 min) and held at 165 °C (20 min). The following isotopes were analyzed: $^{23}$Na, $^{24}$Mg, $^{31}$P, $^{39}$K, $^{44}$Ca, 51V, $^{52}$Cr, $^{55}$Mn, $^{56}$Fe, $^{59}$Co, $^{63}$Cu, $^{66}$Zn, $^{78}$Se, $^{95}$Mo, and $^{111}$Cd by Inductively Coupled Mass Spectrometry (ICP-MS) using an Agilent 7700X spectrometer instrument (Agilent Technologies), that was equipped with a MicroMist nebulizer and nickel sampler and skimmer cones. Erbium was used as an internal standard. Data acquisition was one point, five replicates, and 100 sweeps per replicate.

### 2.4. Amino Acid Analysis

The samples were prepared using standardized AOAC Official Method 985.28, 2021 [22]. Acid hydrolysis was used to prepare the samples for the analysis of histidine (His), serine (Ser), arginine (Arg), glycine (Gly), aspartic acid (Asp), glutamic acid (Glu), threonine (Thr), alanine (Ala), proline (Pro), lysine (Lys), tyrosine (Tyr), valine (Val), Isoleucine (ILeu), leucine (Leu), and phenylalanine (Phe). For the analysis of methionine (Met) and cysteine (Cys), the oxidation procedure was used.

Analysis of the amino acids used pre-column AccQ-Tag derivatization and the Waters Acquity (ultra-pressure liquid chromatography) UPLC application [23] that was published previously [24]. This system, in combination with the AccQ-Tag Ultra method enables the derivatization of amino acids and separation of the derivatives using reverse phase UPLC with the derivatives being identified according to their ultraviolet (UV) absorbance. For this analysis, an Accq-Tag ULTRA C18 1.7 μm 2.1 × 100 mm column was used on an Acquity UPLC system that was equipped with a TUV detector (260 nm), a quaternary solvent system (0.7 mL·min$^{-1}$), and a column oven (20 °C) over 10 min for the separation.

### 2.5. Phytochemical Analysis

To measure the anthocyanin content, the buckwheat samples were extracted and hydrolyzed using an adapted method from Zhang et al. 2004 [25] as described in Neacsu et al. 2021 [24]. Briefly, the samples (approximately 0.1 g dry weight; n = 3) were mixed with the extraction mixture (methanol: water: hydrochloric acid; 50:33:17; *v:v:v*) and hydrolyzed at 100 °C for 60 min followed by HPLC analysis.

The method that was used to measure the flavonoid content in the buckwheat samples was published previously [24] Briefly, the samples (0.1 g dry weight; n = 3) were suspended in hydrochloric acid (0.2 M) and extracted with ethyl acetate. The solvent was then removed, and the extracted material hydrolyzed with hydrochloric acid (1 M) for one hour at 90 °C. After the hydrolysis, the samples were extracted again into ethyl acetate and after the solvent evaporation, the residue was dissolved in methanol and analyzed by LC-MS (detailed below).

The method that was used to measure the phenolic acid content in the buckwheat samples was published previously [26]. Briefly, the samples (approximately 0.1 g dry weight; n = 3) were suspended in hydrochloric acid (0.2 M) and extracted into ethyl acetate. The resulting pellet was then hydrolyzed using alkaline (4 M) and acid (6 N) conditions. The phenolics that were extracted in this three-step extraction process were measured using LC-MS (detailed below).

To prepare the samples for LCMS analysis, Internal Standard 1 for negative mode mass spectrometry (IS1; $^{13}$C benzoic acid), and Internal Standard 2 for positive mode mass spectrometry (IS2; 2-amino-3,4,7,8-tetramethylimidazo [4,5-f] quinoxaline) was added to an aliquot of the extracts that were prepared as described above identical as published previously [24].

The LC–MS analysis methods that were used have been published previously [26,27]. For the liquid chromatography separation of the plant metabolites, an Agilent 1100 LC–MS system (Agilent Technologies, Wokingham, UK) using a Zorbax Eclipse

5 µm, 150 mm × 4.6 mm C18 column (Agilent Technologies) was used. There were three solvent gradients that were used for the separation of the different categories of metabolites. The mobile phase solvents were water containing 0.1% acetic acid (A) and acetonitrile containing 0.1% acetic acid (B). The LC eluent was directed into, without splitting, an ABI 3200 triple quadrupole mass spectrometer (Applied Biosystems; Warrington, UK) that was fitted with a Turbo Ion Spray™ (TIS) source. All the metabolites were quantified using multiple reaction monitoring (MRM). For all the phytochemical quantifications, the standard calibrations were over a concentration interval of 2 to 10 pg/µL. The threshold that was used for the quantification was a signal to noise ratio of 3 to 1. All the ion transitions for each of the metabolites were determined based upon their molecular ions and strong fragment ions; their voltage parameters, declustering potential, collision energy, and cell entrance/exit potentials were optimized individually for each metabolite and have been previously described [17,26].

The quantification of the anthocyanidins was performed as described previously [24] using a 1260 Infinity HPLC from Agilent (Wokingham, UK) and a Synergi 4 µm Polar-RP 80A (250 × 4.6 mm) column with a Polar-RP 4 × 3 mm pre-column from Phenomenex (Macclesfield, UK) that was equipped with a diode array detection (DAD). The spectra were recorded between 200 and 700 nm and the chromatograms were monitored at 530 nm. The HPLC solvent A was formic acid (2.125%) and B acetonitrile/methanol (85:15, *v/v*). The HPLC method for anthocyanidin analysis was adapted from Zhang et al. (2004) [25]. The solvent program was isocratic 18% B with a constant flow (1 mL/min) at 35 °C.

### 2.6. Physical Properties

The moisture content was determined gravimetrically by placing approximately 1.5 g (exact weight recorded) in a pre-weighed crucible and drying at 100 °C in an oven overnight until a constant weight was reached (approximately 22 h). The crucibles were then cooled in a desiccator and reweighed. The loss of weight on drying is used to calculate the amount of moisture in powder by using the formula:

$$Moisture\ content\ (\%) = \frac{Weight\ of\ water\ loss}{Weight\ of\ powder} \times 100 \tag{1}$$

To determine water solubility, approximately 1 g of flour (exact weight recorded) was suspended onto the surface of 100 g of Milli-Q® water at 25 °C in 500 mL-beaker (diameter 80 mm). The mixture was then stirred continuously at 800 rpm for 7 min using a Cole Parmer digital stirrer. The samples were allowed to stand for 30 s before transferring 20 mL of the mixture into a centrifuge tube and were then centrifuged for 10 min at 2900× *g*. Approximately 8–12 g of the supernatants were weighed and placed into a dry pre-weighed crucible and were allowed to dry overnight to a constant weight at 100 °C. The crucibles were cooled in a desiccator and reweighed. The solubility of the powders was calculated using the following equation:

$$Solubility\ (\%) = \frac{(100 + a) \times \%TS}{a \times (100 - b)/100} \tag{2}$$

where *a* is the amount of flour (g), *b* is moisture content in the flour, and % *TS* is the percentage of dry matter in the supernatant.

Dispersibility was determined by adding 1 g of powder (exact weight recorded) onto the surface of 100 g of Milli-Q® water at 25 °C in 500 mL-beaker (diameter 80 mm). The mixture was stirred continuously at 800 rpm for 7 min (Cole Parmer digital stirrer), it was allowed to stand for 30 s, and then 20 mL of the mixture were transferred through a 60-mesh (~210 µ) sieve. Approximately 8–10 g of the filtrates were then transferred into a dry crucible and were allowed to dry overnight to a constant weight at 100 °C. The crucibles

were cooled in a desiccator and were reweighed. Dispersibility was calculated using the following equation:

$$Dispersibility\ (\%) = \frac{(100 + a) \times \%TS}{a \times (100 - b)/100} \tag{3}$$

where $a$ is the amount of flour (g), $b$ is moisture content in the flour, and $\% TS$ is the percentage of dry matter in the supernatant.

Wettability was determined by the static wetting test with some modifications [28]. Briefly, 1 g of flour (exact weight recorded) was sprinkled onto the surface of 100 g of Milli-Q® water at 25 °C in 500 mL-beaker (diameter 80 mm). Wettability is determined by visual observation and expressed as the time (seconds) that is required for the flour to sink into the water.

To determine the loose bulk density, approximately 2 g (exact weight recorded) of flour was gently transferred into a 10 mL measuring cylinder (without shaking or tapping the cylinder). The volume of the flour was recorded and used in a calculation of the loose bulk density. For tapped density determination, the above procedure was modified by tapping the cylinder on a rubber mat from a height 15 cm for 120 times or until constant volume was achieved and the volume of flour was recorded. The loose and tapped bulk densities of the flours were calculated by using the following equations:

$$Loose\ bulk\ density = weight\ of\ flour\ (g)/bulk\ powdered\ volume\ (cm^3) \tag{4}$$

$$Tapped\ bulk\ density = weight\ of\ flour\ (g)/tapped\ powdered\ volume\ (cm^3) \tag{5}$$

The flour flowability and compressibility were evaluated in terms of the Hausner ratio (HR) and the Carr's compressibility index (CI), respectively. HR and CI (%) were calculated from the loose and bulk densities as follows:

$$HR = \frac{\rho\ tapped}{\rho\ bulk} \tag{6}$$

$$CI = \frac{(\rho\ tapped - \rho\ bulk)}{\rho\ tapped} \times 100 \tag{7}$$

The flowability chart (Table 1) classifies the values of the Carr index and Hausner ratio in terms of the flow property and provides an indicative range of the quality of the flour.

**Table 1.** Classification of flowability and cohesiveness of flours based on the Carr index and Hausner ratio [29,30].

| Flowability | Carr Index (%) | Cohesiveness | Hausner Ratio |
|---|---|---|---|
| Very good | <15 | Low | <1.2 |
| Good | 15–20 | Intermediate | 1.2–1.4 |
| Fair | 20–35 | High | >1.4 |
| Bad | 35–45 | | |
| Very bad | >45 | | |

The color properties were determined by using a Konica Minolta CR1 10 color meter (Konica Minolta Solutions Ltd., Basildon, UK). The measurements were made using the International Commission on Illumination (CIE) L* (lightness), a* (redness to bluishness), b* (yellowness to greenness) system.

### 2.7. Statistical Analysis

All the nutrient and phytochemical data were averaged from three technical replicates of the samples and are reported as means ± standard deviations. The differences between the concentrations of various macronutrients, micronutrients, and phytochemicals that were measured in the buckwheat samples in this study were assessed using a one-way analysis

of variance (ANOVA) that was conducted in version 28 of IBM SPSS (SPSS for Windows 22, SPSS Inc., Chicago, IL, USA) to identify the differences among the means by the Bonferroni post hoc test. Statistical significance was set at $p < 0.05$. The plant metabolites (non-nutrients) from the buckwheat samples were analyzed by principal component analysis (PCA), unit variance (UV)-scaled using SIMCA 14.1 (Umetrics, Cambridge, UK). The physical properties data are reported as the means $\pm$ standard errors (SE) from at least triplicate measurements from each flour. Normality of the data was validated with Kolmogorov–Smirnov and Shapiro–Wilks tests. Analysis of variance (ANOVA) was conducted in version 27 of IBM SPSS (SPSS for Windows 22, SPSS Inc., Chicago, IL, USA) to identify differences among means by the Bonferroni post hoc test. Statistical significance was set at $p < 0.05$.

## 3. Results

### 3.1. Macronutrient, Micronutrient, and Phytochemical Composition of the Buckwheat Samples

#### 3.1.1. Macronutrient Composition

The protein (determined as crude nitrogen content), total fat, total carbohydrate, resistant starch, soluble non-starch polysaccharides (NSP), and insoluble NSP content as % of dry weight (including ash and dry matter) for the buckwheat samples from the UK and Brazil (BR) are presented in Table 2. Overall, the ANOVA analysis showed significant differences between the samples for all the macronutrients except for soluble NSP (Table 2).

**Table 2.** Macronutrient content: protein, fat, total carbohydrate, resistant starch, soluble non-starch polysaccharides (NSP), and insoluble NSP content, expressed as % of dry weight $\pm$ SD (n = 3) including ash and dry matter for UK Hulled Buckwheat, Brazil (BR) Hulled Buckwheat and BR Dehulled Buckwheat samples. One-way overall ANOVA $p$-values, where ns stands for non-significant differences and LSD (least significant difference) values.

| Buckwheat Sample | Dry Matter | Ash | Protein | Fat | Carbohydrate Total | Resistant Starch | Soluble NSP | Insoluble NSP |
|---|---|---|---|---|---|---|---|---|
| | **% Dry Weight** | | | | | | | |
| BR Dehulled | 86.43 $\pm$ 0.1 | 1.89 $\pm$ 0.01 | 14.13 $\pm$ 0.50 | 2.30 $\pm$ 0.05 | 69.30 $\pm$ 2.55 | 0.64 $\pm$ 0.00 | 0.06 $\pm$ 0.00 | 1.96 $\pm$ 0.07 |
| BR Hulled | 86.42 $\pm$ 0.02 | 1.88 $\pm$ 0.03 | 11.71 $\pm$ 0.40 | 1.71 $\pm$ 0.07 | 56.52 $\pm$ 1.85 | 0.56 $\pm$ 0.03 | 0.06 $\pm$ 0.03 | 7.80 $\pm$ 0.14 |
| UK Hulled | 83.38 $\pm$ 0.07 | 1.75 $\pm$ 0.03 | 12.24 $\pm$ 0.42 | 1.44 $\pm$ 0.02 | 46.70 $\pm$ 4.43 | 0.38 $\pm$ 0.02 | 0.03 $\pm$ 0.00 | 11.05 $\pm$ 0.25 |
| Overall ANOVA ($p$-values) | <0.001 | <0.001 | 0.001 | <0.001 | <0.001 | <0.001 | ns | <0.001 |
| LSD | 0.1406 | 0.05099 | 0.879 | 0.1001 | 6.27 | 0.04680 | 0.04007 | 0.3212 |

The dehulled buckwheat sample from Brazil was significantly higher ($p < 0.01$) than the other two buckwheat samples in the content of protein 14.13 $\pm$ 0.50%. The UK hulled sample has a significantly lower content of fat, total carbohydrates, and resistant starch in comparison with the other buckwheat samples. The UK hulled sample had a significantly higher content ($p < 0.001$) of insoluble NSP, 11.05 $\pm$ 0.25, in comparison with the other buckwheat samples (Table 2). The LSD (least significant difference) values are presented in Table 2 and a detailed list of ANOVA post hoc Bonferroni analysis $p$-values can be found in the Supplementary Data (Table S1).

The insoluble NSP is the main form of dietary fiber that is present in the buckwheat samples (Table 3). The insoluble NSP is mainly comprised of glucose, xylose, and uronic acids (Table 3), all these monomer units being in different and higher quantities in the hulled buckwheat samples (Table 3).

#### 3.1.2. Microelement Composition

The buckwheat samples were a rich source of several minerals. Overall, ANOVA analysis showed significant differences between the samples for all the microelements except for sodium and potassium (Table 4). The dehulled Brazilian sample was significantly higher in magnesium and phosphorus than the other buckwheat samples. The UK hulled sample was significantly higher in calcium, magnesium ($p < 0.001$), and zinc ($p < 0.001$)

than the other buckwheat samples (Table 4). The LSD (least significant difference) values are presented in Table 4 and a detailed list of ANOVA post hoc Bonferroni analysis *p* values can be found in the Supplementary Data (Table S1).

**Table 3.** Monosaccharide composition of the soluble non-starch polysaccharides (NSP) and insoluble NSP content, expressed as % of dry weight ± SD (n = 3) for UK Hulled Buckwheat, Brazil (BR) Hulled Buckwheat and BR Dehulled Buckwheat samples.

| Buckwheat Samples | | Rhamnose | Fucose | Arabinose | Monosaccharides (% Dry Weight) | | Galactose | Glucose | Uronic Acid |
| | | | | | Xylose | Mannose | | | |
|---|---|---|---|---|---|---|---|---|---|
| BR Dehulled | soluble NSP | 0 ± 0 | 0 ± 0 | 0.001 ± 0.0016 | 0.003 ± 0.0044 | 0 ± 0 | 0 ± 0 | 0.004 ± 0.0032 | 0.052 ± 0.0054 |
| | insoluble NSP | 0.068 ± 0.0138 | 0.028 ± 0.0049 | 0.39 ± 0.005 | 0.133 ± 0.0093 | 0.026 ± 0.0018 | 0.107 ± 0.0041 | 0.656 ± 0.0051 | 0.549 ± 0.0406 |
| BR Hulled | soluble NSP | 0 ± 0 | 0.002 ± 0.0043 | 0.002 ± 0.0032 | 0 ± 0.0005 | 0.002 ± 0.0037 | 0.002 ± 0.0029 | 0.002 ± 0.0043 | 0.048 ± 0.0154 |
| | insoluble NSP | 0.118 ± 0.0003 | 0.042 ± 0.0159 | 0.448 ± 0.0243 | 1.807 ± 0.0515 | 0.056 ± 0.001 | 0.199 ± 0.0037 | 3.939 ± 0.0677 | 1.192 ± 0.0211 |
| UK Hulled | soluble NSP | 0 ± 0 | 0 ± 0 | 0 ± 0 | 0 ± 0 | 0 ± 0 | 0 ± 0 | 0 ± 0 | 0.034 ± 0.0037 |
| | insoluble NSP | 0.139 ± 0.0039 | 0.04 ± 0.0018 | 0.701 ± 0.0101 | 3.499 ± 0.064 | 0.091 ± 0.0021 | 0.251 ± 0.0025 | 4.704 ± 0.0623 | 1.629 ± 0.1312 |

**Table 4.** The main microelements: Na (sodium), Mg (magnesium), P (phosphorus), K (potassium), Ca (calcium), Mn (manganese), Fe (iron), Cu (cupper), Zn (zinc), and Mo (molybdenum) expressed in mg/Kg ± SD (n = 3) dry weight of UK Hulled Buckwheat, Brazil (BR) Hulled Buckwheat and BR Dehulled Buckwheat samples that were obtained by quantitative ICP-MS analysis. One-way overall ANOVA *p*-values, where ns stands for non-significant differences and LSD (least significant difference) values.

| mg/kg Dry Weight | BR Dehulled Buckwheat | BR Hulled Buckwheat | UK Hulled Buckwheat | Overall ANOVA (*p*-Values) | LSD |
|---|---|---|---|---|---|
| Na | 14.36 ± 8.46 | 8.91 ± 14.4 | 24.02 ± 2.93 | ns | 19.56 |
| Mg | 2573.5 ± 55.38 | 2079.8 ± 158.58 | 2203.22 ± 89.8 | <0.01 | 219.7 |
| P | 4384.1 ± 155.58 | 3181.76 ± 235.42 | 3768.07 ± 237.99 | 0.001 | 425.8 |
| K | 4978.26 ± 155.71 | 5076.9 ± 389.68 | 5276.4 ± 319.08 | ns | 608.1 |
| Ca | 117.52 ± 7.64 | 403.52 ± 30.36 | 491.33 ± 24.81 | <0.001 | 46.08 |
| Mn | 13.69 ± 0.55 | 16.42 ± 1.18 | 43.32 ± 1.62 | <0.001 | 2.400 |
| Fe | 24.81 ± 0.66 | 35.63 ± 3.38 | 35.97 ± 1.55 | 0.001 | 4.361 |
| Cu | 5.6 ± 0.14 | 4.52 ± 0.27 | 6.05 ± 0.27 | <0.001 | 0.4723 |
| Zn | 25.32 ± 1.34 | 20.78 ± 1.41 | 34.44 ± 1.53 | <0.001 | 2.851 |
| Mo | 0.41 ± 0.02 | 0.41 ± 0.01 | 0.23 ± 0.01 | <0.001 | 0.03712 |

### 3.1.3. Amino Acid Composition

Overall, the ANOVA analysis for the amino acids content of the buckwheat samples showed significant differences for histidine, arginine, glycine, threonine, tyrosine, leucine, and phenylalanine (Table 5). Although there were no significant differences that were found (ANOVA analysis) between the buckwheat samples for the rest of the amino acids that were measured, the dehulled samples from Brazil had higher values except for serine (Table 5). The LSD (least significant difference) values are presented in Table 5 and a detailed list of ANOVA post hoc Bonferroni analysis *p*-values can be found in the Supplementary Data (Table S1).

**Table 5.** The amino acid composition in µmoles/g dried sample ± SD (n = 3) for UK Hulled Buckwheat, Brazil (BR) Hulled Buckwheat and BR Dehulled Buckwheat samples. One-way overall ANOVA *p*-values, where ns stands for non-significant differences and LSD (least significant difference) values.

| µmoles/g Sample | BR Dehulled Buckwheat | BR Hulled Buckwheat | UK Hulled Buckwheat | Overall ANOVA (*p*-Values) | LSD |
|---|---|---|---|---|---|
| His | 34.36 ± 1.77 | 31.7 ± 1.38 | 35.66 ± 3.11 | 0.01 | 2.593 |
| Ser | 101.53 ± 3.58 | 92.99 ± 2.83 | 101.97 ± 5.2 | ns | 7.31 |
| Arg | 129.34 ± 6.27 | 113.93 ± 4.04 | 131.42 ± 9.31 | <0.01 | 8.62 |

**Table 5.** *Cont.*

| μmoles/g Sample | BR Dehulled Buckwheat | BR Hulled Buckwheat | UK Hulled Buckwheat | Overall ANOVA (*p*-Values) | LSD |
|---|---|---|---|---|---|
| Gly | 168.16 ± 7.6 | 154.01 ± 5.12 | 169.65 ± 7.02 | <0.05 | 11.76 |
| Asp | 94.43 ± 4.89 | 85.09 ± 2.42 | 88.14 ± 6.89 | ns | 7.65 |
| Glu + Gln | 156.9 ± 9.37 | 142.32 ± 7.96 | 146.98 ± 9.91 | ns | 14.46 |
| Thr | 64.74 ± 2.07 | 59.84 ± 1.34 | 64.7 ± 6.3 | 0.01 | 3.463 |
| Ala | 72.32 ± 3.12 | 67.27 ± 3.28 | 69.41 ± 4.2 | ns | 5.799 |
| Pro | 58.58 ± 2.38 | 54.87 ± 2.18 | 57.93 ± 3.37 | ns | 4.031 |
| Lys | 51.13 ± 2.05 | 47.86 ± 3.87 | 46.99 ± 4.19 | ns | 5.317 |
| Tyr | 32.39 ± 1.68 | 27.5 ± 1.49 | 32.76 ± 1.76 | <0.01 | 2.936 |
| Val | 68.07 ± 2.43 | 63.66 ± 3.91 | 63.6 ± 12.64 | ns | 5.666 |
| Ileu | 45.1 ± 1.71 | 42.12 ± 2.6 | 42.47 ± 8.6 | ns | 3.827 |
| Leu | 89.42 ± 2.96 | 82.03 ± 2.98 | 87.15 ± 8.14 | <0.05 | 5.522 |
| Phe | 64.37 ± 3.74 | 58.08 ± 2.95 | 67.29 ± 6.3 | <0.01 | 5.631 |
| Cys | 59.7 ± 3.3 | 51.85 ± 4.61 | 50.84 ± 4.68 | ns | 8.48 |
| Met | 21.54 ± 1.4 | 18.36 ± 1.66 | 17.9 ± 1.62 | ns | 3.132 |

Where: histidine (His), serine (Ser), arginine (Arg), glycine (Gly), aspartic acid (Asp), glutamic acid and glutamine (Glu + Gln), threonine (Thr), alanine (Ala), proline (Pro), lysine (Lys), tyrosine (Tyr), valine (Val), Isoleucine (ILeu), leucine (Leu), phenylalanine (Phe), methionine (Met), and cysteine (Cys).

3.1.4. Phytochemical Composition

Principal component analysis (PCA) of the plant metabolites that were measured (196 molecules) presented a cluster for the hulled buckwheat samples from both Brazil and the UK and a clear delimitation of the dehulled sample from the UK (Figure 1). This is as expected, as 22 phenolic molecules were significantly different between the hulled and dehulled samples from Brazil and only four phenolics were significantly different between the hulled samples. Anthocyanins were the most abundant phenolics that were measured in the buckwheat samples, overall pelargonidin ($p < 0.05$) and cyanidin ($p < 0.01$) being significantly different among the buckwheat samples (ANOVA analysis) (Table 6 and Figure 1). Catechins were only found in the hulled buckwheat samples (Table 6) and from the most abundant phenolics that were measured (concentrations higher than 5 mg/kg), only benzoic-, 2,5-hydroxybenzoic-, and p-hydroxybenzoic acid were found in non-significantly different amounts ($p < 0.001$) among buckwheat samples (Table 6 and Figure 1). Overall, the ANOVA *p*-values only for significant differences ($p < 0.05$), including the LSD value for the most abundant plant metabolites (with concentrations higher than 5 mg/kg) are presented in Table 6 and a detailed list of ANOVA post hoc Bonferroni analysis *p*-values for the most abundant phenolics that were measured in buckwheat samples can be found in the Supplementary Data (Table S1).

*3.2. Physical Properties of Buckwheat Samples*

Moisture levels in powdered foods should be low (ideally <6%) to inhibit or delay the agglomeration of wet particles during storage and to prevent caking, which can adversely affect the quality and acceptability of certain products [31]. The moisture content in the buckwheat samples was significantly different with the UK buckwheat sample being significantly lower ($p < 0.05$) than the other buckwheat samples. This suggests suitability for long-term storage (Table 7).

Bulk density is an important property for powdered samples from an economical perspective. The bulk density is inversely proportional to packaging volume and as a result increasing this physical property is desirable to reduce packaging and transportation costs [32]. Loose and bulk density values of the UK buckwheat flour sample was significantly lower ($p < 0.05$) in comparison with the other samples (Table 7).

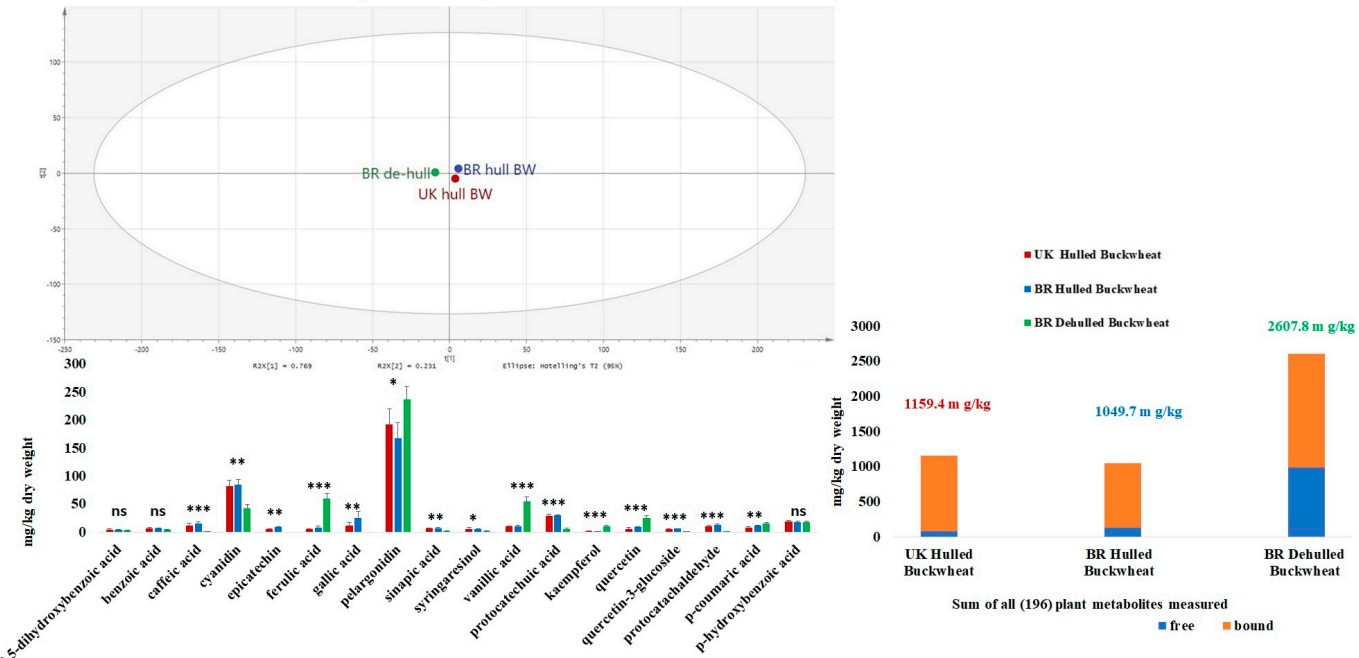

**Figure 1.** Principal component analysis (PCA), unit variance (UV) scaled plots for the first two PCs of all the plant metabolites that were detected in the buckwheat samples (upper left). The most abundant phenolics in buckwheat with concentrations higher than 5 mg/kg dry weight that were measured in mg/dry flour ± SD, n = 3 (lower left). The total amount (mg/kg dry weight) as sum of 196 plant metabolites that were measured, with free and bound labile fractions color coded (right). Where ns stands for non-significant, * for *p* < 0.05, ** for *p* < 0.01, and *** for *p* < 0.001 that were obtained from one way ANOVA analysis.

**Table 6.** Plant metabolites in the buckwheat samples. Concentration (mg/kg dry weight) ± SD (n = 3) of benzoic acids, benzaldehydes, benzenes, acetophenones, cinnamic acids, phenylpropionic acids, phenylacetic acids, phenylpyruvic acids, phenyllactic acids, quinadilic acid, coniferyl alcohol, 4-hydroxy-3-methoxy cinnamyl alcohol, 4-methylcatechol, flavonoids, isoflavonoids, catechins, lignans, and anthocyanins in buckwheat. Overall, the ANOVA *p*-values only for significant differences < 0.05 (with LSD value) among the most abundant plant metabolites (with concentrations higher than 5 mg/kg).

| Plant Metabolite | BR Dehulled Buckwheat | BR Hulled Buckwheat | UK Hulled Buckwheat | Overall ANOVA *p*-Values When <0.05 (with LSD Value) |
|---|---|---|---|---|
| | **mg/kg Dry Weight** | | | |
| *Benzoic acid* | 4.281 ± 0.798 | 5.832 ± 1.338 | 6.467 ± 1.504 | <0.001 (1.649) |
| Salicylic acid | 0.622 ± 0.197 | 3.381 ± 0.574 | 4.082 ± 0.744 | |
| *m*-Hydroxybenzoic acid | nd | nd | nd | |
| *p*-Hydroxybenzoic acid | 17.194 ± 2.509 | 17.291 ± 1.786 | 18.661 ± 1.93 | |
| *2,3-Dihydroxybenzoic acid* | 0.029 ± 0.003 | 2.682 ± 0.465 | 3.488 ± 0.8 | <0.001 (1.089) |
| 2,4-Dihydroxybenzoic acid | 0.189 ± 0.037 | 0.13 ± 0.008 | 0.133 ± 0.063 | |
| 2,5-Dihydroxybenzoic acid | 2.728 ± 0.356 | 4.311 ± 0.604 | 4.296 ± 0.996 | |
| 2,6-Dihydroxybenzoic acid | 0.18 ± 0.018 | 0.074 ± 0.01 | 0.088 ± 0.029 | |
| *Protocatechuic acid* | 5.3 ± 1.344 | 29.545 ± 1.453 | 28.381 ± 3.349 | <0.001 (4.142) |
| 3,5-Dihydroxybenzoic acid | nd | nd | nd | |
| *o*-Anisic acid | nd | nd | nd | |
| *m*-Anisic acid | nd | 0.109 ± 0.007 | 0.113 ± 0.012 | |
| *p*-Anisic acid | 0.522 ± 0.106 | 0.833 ± 0.124 | 0.967 ± 0.158 | |

**Table 6.** *Cont.*

| Plant Metabolite | BR Dehulled Buckwheat | BR Hulled Buckwheat | UK Hulled Buckwheat | Overall ANOVA *p*-Values When <0.05 (with LSD Value) |
|---|---|---|---|---|
| | mg/kg Dry Weight | | | |
| *Gallic acid* | nd | 24.877 ± 11.394 | 11.55 ± 5.095 | 0.01 (12.90) |
| *Vanillic acid* | 54.471 ± 8.46 | 10.68 ± 1.664 | 9.924 ± 1.1 | <0.001 (3.075) |
| Syringic acid | 0.934 ± 0.152 | 4.003 ± 3.221 | 2.405 ± 0.309 | |
| 3,4-Dimethoxybenzoic acid | 0.299 ± 0.043 | 0.401 ± 0.046 | 0.37 ± 0.044 | |
| p-Hydroxybenzaldehyde | 1.686 ± 0.37 | 3.246 ± 0.364 | 3.794 ± 0.604 | |
| *Protocatechualdehyde* | 0.18 ± 0.048 | 12.446 ± 2.119 | 10.47 ± 1.494 | <0.001 (2.067) |
| 3,4,5-Trihydroxybenzaldehyde | nd | nd | nd | |
| *Vanillin* | 0.977 ± 0.297 | 3.603 ± 0.747 | 3.906 ± 1.026 | <0.01 (1.425) |
| Isovanillin | nd | nd | nd | |
| Syringin | 0.286 ± 0.054 | 1.225 ± 0.157 | 1.523 ± 0.253 | |
| 3-Methoxybenzaldehyde | nd | nd | nd | |
| 3,4-Dimethoxybenzaldehyde | nd | 0.01 ± 0.002 | nd | |
| 3,4,5-Trimethoxybenzaldehyde | nd | nd | nd | |
| Cinnamic acid | 1.311 ± 0.246 | 1.038 ± 0.131 | 0.649 ± 0.112 | |
| *o*-Coumaric acid | nd | nd | nd | |
| *m*-Coumaric acid | nd | nd | nd | |
| *p-Coumaric acid* | 14.846 ± 2.573 | 11.03 ± 1.037 | 8.033 ± 0.906 | <0.01 (2.525) |
| *Caffeic acid* | 0.562 ± 0.12 | 15.223 ± 2.451 | 11.48 ± 3.624 | <0.001 (4.290) |
| *Ferulic acid* | 59.91 ± 9.101 | 7.496 ± 3.042 | 5.649 ± 0.618 | <0.001 (8.65) |
| *Sinapic acid* | 1.974 ± 0.49 | 6.57 ± 1.758 | 6.088 ± 0.98 | <0.01 (2.290) |
| 3-Methoxycinnamic acid | nd | nd | nd | |
| 4-Methoxycinnamic acid | nd | nd | nd | |
| 3,4-Dimethoxycinnamic acid | nd | 0.096 ± 0.004 | 0.11 ± 0.014 | |
| 3,4,5-Trimethoxycinnamic acid | nd | 0.942 ± 0.035 | 0.949 ± 0.134 | |
| Phenyl propionic acid | nd | nd | nd | |
| 2-Hydroxyphenylpropionic acid | nd | nd | nd | |
| 3-Hydroxyphenylpropionic acid | nd | nd | nd | |
| 4-Hydroxyphenylpropionic acid | nd | nd | nd | |
| 3,4-Dihydroxyphenylpropionic acid | nd | nd | nd | |
| 4-Hydroxy-3-methoxyphenylpropionic acid | 0.292 ± 0.071 | 0.187 ± 0.051 | 0.269 ± 0.083 | |
| 3-Methoxyphenylpropionic acid | nd | nd | nd | |
| Phenol | nd | 32.366 ± 6.472 | 31.267 ± 3.407 | |
| 1,2-Hydroxybenzene | nd | 0.066 ± 0.059 | nd | |
| 1,3-Hydroxybenzene | nd | nd | nd | |
| 1,2,3-Trihydroxybenzene | nd | nd | nd | |
| 4-Hydroxyacetophenone | 0.307 ± 0.038 | 0.158 ± 0.019 | 0.173 ± 0.015 | |
| 4-Hydroxy-3-methoxyacetophenone | 0.28 ± 0.042 | 0.259 ± 0.056 | 0.246 ± 0.035 | |
| 4-Hydroxy-3,5-dimethoxyacetophenone | nd | 0.087 ± 0.01 | nd | |
| 3,4-Dimethoxyacetophenone | nd | nd | nd | |
| 3,4,5-Trimethoxyacetophenone | nd | nd | nd | |
| Phenylacetic acid | 0.472 ± 0.224 | 1.809 ± 0.505 | 1.113 ± 0.216 | |
| 3-Hydroxyphenylacetic acid | nd | nd | nd | |
| 4-Hydroxyphenylacetic acid | 0.914 ± 0.36 | 38.692 ± 5.791 | 28.872 ± 9.651 | |
| 3,4-Dihydroxyphenylacetic acid | nd | 0.262 ± 0.228 | 0.221 ± 0.214 | |
| 4-Hydroxy-3-methoxyphenylacetic acid | nd | nd | nd | |
| 4-Methoxyphenylacetic acid | nd | nd | nd | |
| Mandelic acid | nd | nd | nd | |
| 3-Hydroxymandelic acid | nd | nd | nd | |
| 4-Hydroxymandelic acid | nd | nd | nd | |
| 3,4-Dihydroxymandelic acid | 0.118 ± 0.055 | 0.49 ± 0.117 | 0.4 ± 0.082 | |
| 4-Hydroxy-3-methoxymandelic acid | nd | nd | nd | |
| Phenyl pyruvic acid | 0.415 ± 0.119 | 0.126 ± 0.02 | 0.118 ± 0.082 | |
| 4-Hydroxyphenylpyruvic acid | 25.02 ± 4.931 | 24.769 ± 2.597 | 23.232 ± 4.753 | |
| Phenyl lactic acid | 0.433 ± 0.047 | 1.474 ± 0.195 | 1.088 ± 0.215 | |
| 4-hydroxyphenyllactic acid | 0.479 ± 0.144 | 1.758 ± 0.23 | 1.473 ± 0.278 | |

**Table 6.** *Cont.*

| Plant Metabolite | BR Dehulled Buckwheat | BR Hulled Buckwheat | UK Hulled Buckwheat | Overall ANOVA *p*-Values When <0.05 (with LSD Value) |
|---|---|---|---|---|
| | mg/kg Dry Weight | | | |
| Anthranilic acid | 0.296 ± 0.036 | 0.276 ± 0.028 | 0.265 ± 0.038 | |
| Quinaldinic acid | nd | nd | nd | |
| Chlorogenic acid | 0.137 ± 0.038 | 0.744 ± 0.642 | 0.137 ± 0.031 | |
| *o*-Hydroxyhippuric acid | nd | nd | nd | |
| Ethylferulate | nd | nd | nd | |
| Coniferyl alcohol | 0.46 ± 0.104 | nd | 0.218 ± 0.068 | |
| *p*-Cresol | nd | nd | nd | |
| 4-Ethylphenol | nd | nd | nd | |
| 4-Methylcatechol | nd | 0.075 ± 0.01 | 0.075 ± 0.015 | |
| Ellagic acid | nd | nd | nd | |
| Ferulic dimer (5-5 linked) | 1.375 ± 0.237 | 0.065 ± 0.045 | 0.043 ± 0.011 | |
| Ferulic dimer (8-8 linked) | nd | nd | nd | |
| Ferulic dimer (8-5 linked) | 3.297 ± 0.372 | 0.079 ± 0.065 | 0.051 ± 0.004 | |
| Ferulic dimer (5-5 hydrogenated) | nd | nd | nd | |
| Resveratrol | nd | nd | nd | |
| Indole | 0.187 ± 0.026 | 0.145 ± 0.013 | 0.168 ± 0.016 | |
| Indole-3-acetic acid | 0.097 ± 0.004 | nd | nd | |
| Indole-3-acrylic acid | nd | nd | nd | |
| Indole-3-propionic acid | nd | nd | nd | |
| Indole-3-carbinol | nd | nd | nd | |
| Indole-3-carboxylic acid | 0.283 ± 0.052 | 0.172 ± 0.01 | 0.2 ± 0.031 | |
| Indole-3-pyruvic acid | 2049.461 ± 149.169 | 474.242 ± 121.003 | 611.974 ± 112.205 | |
| Indole-3-methyl | nd | nd | nd | |
| Indoe-3-lactic acid | 0.025 ± 0.001 | 0.024 ± 0.005 | nd | |
| Coumarin | 0.007 ± 0.006 | 0.018 ± 0 | 0.003 ± 0.006 | |
| Psoralen | nd | nd | nd | |
| 8-Methylpsoralen | 0.002 ± 0.002 | 0.001 ± 0.001 | nd | |
| Bergapten | 0.006 ± 0.002 | 0.01 ± 0.008 | 0.001 ± 0.001 | |
| Tangeretin | 0.041 ± 0.016 | 0.074 ± 0.07 | 0.007 ± 0.001 | |
| Coumesterol | nd | nd | nd | |
| *Catechin* | nd | 2.8 ± 0.233 | 1.254 ± 0.301 | <0.001 (0.877) |
| *Epicatechin* | nd | 9.389 ± 0.347 | 5.698 ± 0.674 | <0.01 (3.848) |
| Gallocatechin | nd | nd | nd | |
| Epigallocatechin | nd | nd | nd | |
| Epigallocatechin gallate | nd | nd | nd | |
| Imperatorin | 0.006 ± 0.002 | 0.013 ± 0.015 | nd | |
| 4-Methylumbelliferone | nd | nd | nd | |
| 7-Hydroxy-4-methyl coumarin | nd | nd | nd | |
| 4-Hydroxy-6-methyl coumarin | nd | nd | nd | |
| Luteolinidin | nd | 0.361 ± 0.099 | 0.244 ± 0.074 | |
| Glycitein | 0.004 ± 0 | 0.002 ± 0.004 | nd | |
| 2-Hydroxybenzyl alcohol | nd | nd | nd | |
| 4-Hydroxy-3-methoxycinnamyl alcohol | nd | nd | nd | |
| Secoisolariciresinol | nd | 0.147 ± 0.02 | nd | |
| Matairesinol | nd | nd | nd | |
| Enterodiol | nd | nd | nd | |
| Enterolactone | nd | nd | nd | |
| *Syringaresinol* | 1.678 ± 0.449 | 4.881 ± 0.774 | 5.657 ± 3.102 | <0.05 |
| Pinoresinol | nd | 0.211 ± 0.043 | 0.224 ± 0.031 | |
| Lariciresinol | nd | nd | nd | |
| Hydroxymatairesinol | nd | nd | nd | |
| 3-Indoleacetonitrile | nd | nd | nd | |
| Indole-3-carboxaldehyde | 0.282 ± 0.039 | 0.217 ± 0.015 | 0.252 ± 0.037 | |

**Table 6.** *Cont.*

| Plant Metabolite | BR Dehulled Buckwheat | BR Hulled Buckwheat | UK Hulled Buckwheat | Overall ANOVA *p*-Values When <0.05 (with LSD Value) |
|---|---|---|---|---|
| | mg/kg Dry Weight | | | |
| *Kaempferol* | 10.069 ± 1.782 | 0.944 ± 0.155 | 1.107 ± 0.451 | <0.001 (1.982) |
| *Quercetin* | 25.574 ± 4.009 | 9.206 ± 0.8 | 5.586 ± 2.495 | <0.001 (5.353) |
| Isoliquiritigenin | nd | 0.01 ± 0 | 0.01 ± 0.001 | |
| Phloretin | nd | 0.049 ± 0.002 | 0.025 ± 0.023 | |
| Eriocitrin | 0.06 ± 0.021 | nd | nd | |
| Naringenin | 0.094 ± 0.013 | 0.158 ± 0.016 | 0.092 ± 0.067 | |
| Naringin | nd | nd | nd | |
| Hesperitin | nd | nd | nd | |
| Morin | nd | nd | nd | |
| Myricetin | nd | nd | nd | |
| *Quercetin-3-glucoside* | 0.537 ± 0.068 | 5.807 ± 0.503 | 4.732 ± 0.527 | <0.001 (0.344) |
| Taxifolin | 0.052 ± 0.005 | 0.208 ± 0.037 | 0.156 ± 0.062 | |
| Genstein | nd | 0.136 ± 0.086 | nd | |
| Scopoletin | 0.012 ± 0.002 | 0.047 ± 0.006 | 0.047 ± 0.013 | |
| Umbelliferone | nd | 0.053 ± 0.007 | 0.046 ± 0.014 | |
| 7,8-Dihydroxy-6-methyl coumarin | nd | nd | nd | |
| Neohesperidin | nd | nd | nd | |
| Hesperidin | nd | 0.004 ± 0.007 | 0.023 ± 0.004 | |
| Quercitrin | nd | 0.11 ± 0.022 | 0.512 ± 0.05 | |
| Biochanin A | 0.182 ± 0.021 | nd | nd | |
| Poncirin | nd | nd | nd | |
| Didymin | nd | nd | nd | |
| Phloridzin | nd | 0.098 ± 0.007 | 0.101 ± 0.012 | |
| Daidzein | nd | 0.136 ± 0.091 | nd | |
| Galangin | nd | nd | nd | |
| Luteolin | 0.016 ± 0.005 | 0.33 ± 0.017 | 0.202 ± 0.107 | |
| Equol | nd | nd | nd | |
| Fisetin | nd | nd | nd | |
| Neoeriocitrin | nd | nd | nd | |
| Formononetin | 0.094 ± 0.097 | 0.176 ± 0.087 | 0.043 ± 0.058 | |
| Apigenin | 0.004 ± 0.001 | 0.038 ± 0.007 | 0.058 ± 0.028 | |
| Gossypin | nd | nd | nd | |
| Tyrosol | nd | nd | nd | |
| Hydroxytyrosol | 0.006 ± 0.01 | nd | nd | |
| Isorhamnetin | 3.194 ± 0.358 | 0.23 ± 0.088 | 0.115 ± 0.086 | |
| *Cyanidin* | 42.884 ± 5.89 | 84.015 ± 9.672 | 81.463 ± 10.418 | <0.01 (17.75) |
| *Pelargonidin* | 236.219 ± 23.463 | 167.659 ± 27.457 | 191.592 ± 28.467 | 0.05 (53.05) |

Where nd is non determined or below the limit of detection and quantification. The underline italics represent significant different values.

**Table 7.** Physical properties of UK Hulled Buckwheat, Brazil (BR) Hulled Buckwheat and BR Dehulled Buckwheat samples.

| Buckwheat Sample | Moisture (%) | Loose Bulk Density (g/mL) | Tapped Bulk Density (g/mL) | Carr Index | Hausner Ratio | Wettability (sec) | Dispersibility (%) | Solubility (%) | Color Values | | |
|---|---|---|---|---|---|---|---|---|---|---|---|
| | | | | | | | | | L* | a* | b* |
| BR Dehulled | 8.66 ± 0.07 [b] | 0.65 ± 0.00 [c] | 0.77 ± 0.00 [b] | 16.13 ± 0.00 [c] | 1.19 ± 0.00 [c] | 1344 ± 21 [c] | 11.32 ± 0.84 [a] | 4.52 ± 0.87 [b] | 47.4 ± 0.5 [b] | 2.0 ± 0.1 [b] | 11.2 ± 0.0 [c] |
| BR Hulled | 8.59 ± 0.03 [b] | 0.61 ± 0.0 [b] | 0.77 ± 0.00 [b] | 21.21 ± 0.00 [b] | 1.27 ± 0.00 [b] | 620 ± 12 [b] | 16.74 ± 2.21 [a] | 9.09 ± 1.73 [ab] | 38.2 ± 0.1 [a] | 1.9 ± 0.0 [b] | 9.4 ± 0.0 [b] |
| UK Hulled | 7.99 ± 0.06 [a] | 0.58 ± 0.01 [a] | 0.73 ± 0.01 [a] | 20.20 ± 0.20 [a] | 1.25 ± 0.00 [a] | 1514 ± 9 [a] | 12.92 ± 1.69 [a] | 11.53 ± 1.54 [a] | 37.9 ± 0.1 [a] | 2.3 ± 0.0 [a] | 9.7 ± 0.0 [a] |

Means with different superscripts in each column are significantly different (*p* < 0.05), values are mean ± standard deviation (n = 3).

The handling properties (flowability and cohesiveness) of the buckwheat flours were determined by the CI index and Hausner ratio, respectively. The dehulled Brazilian sample was ranked as good in terms of flowability and low for cohesiveness, whereas the Brazilian hulled sample and the UK sample were ranked as fair and intermediate for flowability and cohesiveness, respectively, according to the classification that is presented in Table 1. It is worth noting that flowability rankings were in accordance with the cohesiveness rankings for all the buckwheat samples.

Rehydration is an important quality indicator of powdered products and determines consumer acceptability. Quick and complete sample rehydration is the desirable criterion for most food applications. The process of rehydration involves several steps such as wetting, dispersion, and solubilization (also known as dissolution) which often overlap each other chronologically [33]. Powdered sample rehydration is determined by various test procedures, which aim to assess different reconstitution properties. Wettability refers to the initial contact of the powder surface with water and is time-dependent [34]. The Brazilian hulled buckwheat sample showed the shortest wetting time, followed by the dehulled Brazilian and UK buckwheat samples. Similarly, Brazilian buckwheat hulled, and UK buckwheat samples exhibited increased dispersibility and solubility compared to the dehulled Brazilian buckwheat sample.

The flour color is critical for consumer acceptability and can have a major impact on the sensory properties of the final product. Hulled buckwheat samples in general appear to be significantly darker, while the removal of the hull results in a lighter, yellow color.

## 4. Discussion

### 4.1. Macronutrient and Micronutrient Composition

The protein values that were reported in this study for the buckwheat samples (11.70–12.14%) are similar to value that was reported by other researchers (13.27%) [35] and similar with other pseudo cereals such as quinoa (14.12%) [36] and chia (16.5450%) [36]. Similarly, the carbohydrate, fat, and ash content are comparable with previous studies on buckwheat (72%, 2.36%, and 2.23%, respectively) [35]. The NSP results suggest that the buckwheat dietary fiber is concentrated in the hull; 100 g of hulled buckwheat contributing to up to half of the daily dietary recommendation for fiber (as NSP) [37]. The present study also shows that the majority of the NSP is in an insoluble form with xylose (23.63%), glucose (51.11%), and uronic acid (15.10%) as the main components from the hulled buckwheat from Brazil; and xylose (31.13%), glucose (42.11%), and uronic acid (14.10%) as the main components of the insoluble NSP from the UK sourced hulled buckwheat. These results also suggest that the insoluble NSP is probably of cellulosic type. Wheat bran is currently the main source of fiber in the UK diet [38], with a similar NSP composition 47.4% xylose, 40.9% arabinose, and 8.4% glucose [39]. Although hulled buckwheat has a similar monomeric structure composition to wheat bran, the ratio differs. Therefore, it is important to further explore buckwheat hull fermentability in order to understand and adopt it as a source of fiber in the human diet. Buckwheat hulls could contribute to the boosting and diversification of fiber consumption, especially in western-type diets.

This study results also showed that the buckwheat samples are a complete source of amino acids. The high quality of buckwheat protein could be explained by the high concentrations of essential amino acids, such as lysine, tryptophan, threonine, and the sulphur-containing amino acids [40]. Moreover, previous research showed that plasma branched-chain amino acids (BCAAs) were lower after buckwheat comparison with a meat-based meal [18], this being relevant for the prevention of chronic diseases such as Type 2 diabetes mellitus (T2DM). Animal studies showed that buckwheat protein (i.e., albumin) has suppressive effects on mammalian alpha amylase and on postprandial hyperglycemia [41]. Furthermore, it has been suggested that if buckwheat is consumed as a staple food, that it has beneficial effects on renal dysfunction in patients with Type 2 diabetes [42]. Overall, 100 g of buckwheat could contribute up to 40% of daily RNI for methionine, valine, tyrosine, up to 60% of daily RNI for cysteine, up to 73% for isoleucine up

to 78% for leucine, up to 87% of RNI for phenylalanine and the daily RNI for threonine. In terms of individual amino acid values, the results were similar and in accordance with other research studying buckwheat from Asia (China and Japan) [43–45], where the authors were reporting a variation of the amino acid content with geographical location. In general, the dehulled samples are higher in individual amino acids (in some cases significantly higher) and, respectively, protein content, suggesting that the hull is low in protein. However, this needs to be explored further using a higher number of hulled and dehulled samples.

Overall, all the buckwheat samples were rich in microelements which could contribute to delivering the recommended nutrient intake; 100 g of buckwheat could deliver up to approximately 83% of the daily RNI for Mg, 78% of the daily RNI for P, 40% of the daily RNI for Fe, 48% of the daily RNI for Zn, 50% of the daily RNI for Cu, 88% of the daily RNI for Mo, and the daily RNI for Mn and Cr. Nutritional studies should, therefore, establish the bioavailability of microelements from buckwheat and explore further their potential as functional ingredients used in food fortification to tackle malnutrition. Analysis also suggests that the hulls are a rich source of calcium and iron, however this needs to be explored further using a higher number of hulled and dehulled samples. The differences in the individual microelement content between hulled samples from Brazil and the UK is likely to reflect the soil conditions in which they were grown; with UK samples having overall higher values.

### 4.2. Phytochemical Composition

The dehulled buckwheat sample was higher in phytochemicals (2607.8 mg/kg); calculated as sum of total plant metabolites that were measured (196 molecules) in comparison with the other buckwheat samples. The plant metabolites from the hulled samples were mainly acid and alkali-labile showing that they were bound to other plant components; with 92.76% and 88.03% of the total plant metabolites measured in the UK and Brazil samples, respectively. This suggests that the hulled buckwheat samples could deliver important bioactive phytochemicals later in the gastrointestinal tract when consumed in food. Dehulling the buckwheat resulted in a more equal distribution of the plant metabolites between free and bound fractions with 62.26% of the metabolites that were measured being in the bound fraction, suggesting that dehulled buckwheat samples could deliver important plant metabolites early in the gastrointestinal tract. There was also a ten-fold increase in the concentration of a tryptophan metabolite, indole-3-pyruvic in the dehulled sample; suggesting that the hull could potentially protect the protein (see Table 6). However, these findings need to be explored further using a higher number of hulled and dehulled samples. The phytochemical analysis also revealed that the hulls are a rich source of catechins, protocatechuic acid, and gallic acid, moreover half of the cyanidin was present in the hull when it was compared with the dehulled buckwheat samples. Anthocyanins were the most abundant phenols in all the buckwheat samples, agreeing with previous work [10,46], with pelargonidin being the most abundant molecule in the buckwheat samples. Pelargonidin counteracts hemoglobin glycation, iron release from the heme protein, and iron-mediated oxidative damages, confirming glycated hemoglobin-associated oxidative stress in diabetes [47]. It has also been shown that pelargonidin has the potential to control the formation and progression of tumors, therefore, it could be part of the development of new antitumor drugs and regimens for human colorectal carcinoma [48]. Other flavonoids that were abundant in the buckwheat samples were quercetin and kaempferol. The catechins, quercetin, caffeic acid, and syringic acid and anthocyanins are all reported in abundance in buckwheat and have been reviewed by other researchers [13]. Many comprehensive reviews gathered evidence on the wide range of benefits than flavonoids can have on human health [49–51] including the prevention of metabolic disorders such as T2DM, and cardiovascular disease.

The presence of phenolic compounds such as gallic acid and catechins in buckwheat confer allelopathic potential and buckwheat is reported to be used as a ground cover crop or green manure that may produce inhibitors, which could suppress weeds [52–55]. Several allelochemicals; ferulic acid, quercetin, and caffeic acid that are present in buckwheat

have been reviewed [56], However, other researchers suggest the allelopathic properties of buckwheat are associated with some disturbances of secondary metabolism in the tissues of weed species [57].

*4.3. Physical Properties*

The differences that were observed in moisture levels between the buckwheat samples can be attributed to differences in the proximate composition. Brazilian buckwheat flours have higher total solid contents (Table 2), which is likely to have a limiting effect on the amount of free water that is available for evaporation [58]. Brazilian samples contain significantly higher levels of carbohydrates and significantly lower levels of NSP compared with the UK buckwheat sample. This indicates that the Brazilian samples contain a higher amount of macromolecules which can bind to water, are more hydrophilic, and are able to retain more moisture.

Loose and bulk density values of the Brazilian samples were significantly higher than the UK buckwheat sample. This effect can be associated with the higher total solid and moisture content that was observed in the Brazilian samples. There are several reports which suggest that increasing the total solid content of flours results in an increased bulk density [59]. In addition, flours with high moisture content tend to have a high bulking weight, due to the presence of water, which is considerably denser than the dry solid [60]. This in turn means that bulk material is more easily accommodated between particles, the volume is reduced, and thus bulk density increases [61]. Bulk density can also impact other flour properties such as flowability [62]. The higher loose bulk density of the dehulled Brazilian flour samples may account for its desirable classification with regards to handling properties, even though it contains significantly higher fat than the other two buckwheat samples [63].

Reconstitution properties are determined to a large extent by the surface composition and hydrophobicity of the samples [64]. The poor reconstitution properties of the dehulled Brazilian sample are mainly attributed to the higher fat content. In addition, higher solid and moisture content can adversely affect the reconstitution properties of flours [58,63]. Finally, there are studies which suggest that flour solubility and tapped density are inversely related [60,65]. Thus, although the composition of the dehulled Brazilian buckwheat flour seems to be beneficial for its handling properties, its reconstitution properties are adversely affected.

The results suggest that the process of dehulling is the main contributing factor to the color differences that are observed between the buckwheat samples. However, these observations need to be explored further using a higher number of hulled and dehulled samples.

## 5. Conclusions

This study brings convincing evidence on the potential of buckwheat to contribute to diet diversification and towards meeting daily dietary recommendations for various nutrients. Overall, 100 g of buckwheat samples delivered approximately half of the daily recommendation for NSP, and important quantities of essential amino acids such as methionine, valine, tyrosine, threonine, cysteine, isoleucine, leucine, and phenylalanine. The crop can also contribute towards meeting the daily requirements for iron, zinc, magnesium, phosphorus, manganese, and molybdenum. Furthermore, the distribution of important amounts of bioactive plant secondary metabolites such as anthocyanins, flavonoids, and phenolic acids in acid- and alkaline-labile fractions of the hulled buckwheat samples suggest potentially that the hulls could be an attractive significant source of bioactive molecules for the human gut and perhaps the basis of functional foods development. This aspect, however, needs further investigation.

The physical properties of the buckwheat flour samples were determined by the solid content and proximate composition. The Brazilian samples showed desirable handling and reconstitution properties; suggesting that the hull removal could be beneficial for increasing the flour bulk density, whereas the hulled samples were more easily rehydrated.

Assessing the nutritional and bioactive phytochemical content together with understanding the physical properties of the buckwheat samples provides valuable information for the food industry to develop buckwheat-rich foods and potential functional foods for nutritional therapies. This information can increase demand for the crop and potentially promote its scope for cultivation further, particularly in countries where a western-style diet predominates such as the UK.

**Supplementary Materials:** The following supporting information can be downloaded at: https://www.mdpi.com/article/10.3390/crops2030021/s1, Table S1: List of one-way ANOVA post hoc Bonferroni test analysis p values for the macronutrients, microelements, amino acids and for the phytochemicals results.

**Author Contributions:** Conceptualization, M.N., V.R. and W.R.R.; methodology, M.N., V.R., W.R.R. and S.D.L.S. formal analysis, S.D.L.S., N.J.V., V.R., H.E.H. and G.J.D.; data curation, M.N., S.D.L.S., H.E.H., V.R. and G.J.D.; writing—original draft preparation, M.N. and V.R.; writing—review and editing, M.N., V.R. and W.R.R.; visualization, M.N., V.R. and W.R.R.; supervision, M.N., V.R. and W.R.R. All authors have read and agreed to the published version of the manuscript.

**Funding:** This research was funded by the Scottish Government's Rural and Environment Science and Analytical Services Division (RESAS) as part of the Strategic Research Programme 2016–2021 and Ministério da Educação Coordenação de Aperfeiçoamento de Pessoal de Nível Superior–CAPES, Brazil.

**Institutional Review Board Statement:** Not applicable.

**Informed Consent Statement:** Not applicable.

**Data Availability Statement:** Not applicable.

**Acknowledgments:** The authors would like to thank to Donna Henderson, Lynn Pirie, and Jodie Park, from the Rowett Institute Analytical Department for doing the proximate, amino acid, and ICPMS analysis; and to the funders: Scottish Government's Rural and Environment Science and Analytical Services Division (RESAS) and Ministério da Educação Coordenação de Aperfeiçoamento de Pessoal de Nível Superior Brazil (CAPES).

**Conflicts of Interest:** The authors declare no conflict of interest.

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
