# Peer review of "Nutritional Content, Phytochemical Profiling, and Physical Properties of Buckwheat (Fagopyrum esculentum) Seeds for Promotion of Dietary and Food Ingredient Biodiversity"

_2673-7655, doi:10.3390/crops2030021_

Round 1

Reviewer 1 Report

The paper deals with an interesting topic that fits into this special issue of crops. Also a high number of different analyses has been carried out. However there are a number of issues that need to be revised before publication can be recommended. Introduction need to be shorten and revised. Buckwheat is a pseudocereal which means that the crop is totally different from cereals. The difference is that it is not a grass. But its seeds are used for similar types of food products as cereals. It can never replace the cereals as the yield and quality of products are not similar. However it is more nutritious and can grow on marginal lands. So it is not adeqwat with the start of the introduction discussing cereals. Material and methods needs to be uppdatera on the sampling. If i understod right three samples where purchased and analyzed with all the different analyses. This means that the differences obtained is between three samples and not between brasilian and british samples or between hulled and dehulled samples. To dras such conclusions you would have needed e.g.50 of each type. Also, as they have not been grown on the same field or place, the variation obtained might be due to cultivation conditions or storing conditions of the samples.this, this study can be seen as a screening of three different samples and give an indication of levels of different components in buckwheat. Results, discussion, conclusion and abstract needs revision along these points. 

Author Response

Thank you for reviewing the manuscript and for your suggestions. We consider these are all valuable points which will increase the clarity and value of the manuscript.

We hopefully have addressed all your comments satisfactorily. Please see attached the letter with our responses in red colour at each of your point raised. All the modifications are also highlighted in red colour in the amended manuscript.

Reviewer 2 Report

Dear Authors,

The work submitted for review entitled "Nutritional content, chemical profiling and physical properties 2 of UK and Brazil grown Buckwheat (Fagopyrum esculentum)" is interesting, but could have been better written and interpreted. Here are the detailed notes:

1)     The purpose of the work should be better explained.

2)     The introduction should end with an alternative research hypothesis to the null hypothesis. This hypothesis should be verified later in the paper.

3)     Botanical nomenclature should be standardized. The consumption raw material of buckwheat is "nuts", not "seeds". You have to use the right botanical names. In addition, the expression 'buckwheat flour' should be used instead of the expression 'buckwheat powder'.

4)     In the chapter "Material and methods", there is no information about where the assessed buckwheat raw material came from, in what soil and climatic conditions the buckwheat was grown, what or what were the cultivars? - this is the basic knowledge that the authors should include in this work. It is not enough to write that the raw material was purchased in a store, because then you cannot refer to the geographical origin.

5)     Based on the analysis of variance, the authors should determine the percentage share of individual sources of variability and their interactions, which would allow for a quantitative and not only qualitative assessment of individual sources of variation and to show to what extent the origin of buckwheat affects the physical and chemical properties of buckwheat fruit (nuts) .

6)     In the discussion of the results and in the discussion, there is no detailed discussion of the impact of the origin of the buckwheat raw material on the consumption quality of flour, which is required in relation to the title of the work.

7)     Conclusions should be generalizing and summarizing. In the main conclusion, the authors emphasize that their research was "the first buckwheat nutritional study". Meanwhile, these were only studies of the chemical composition and physical properties of buckwheat nuts. Nutrition studies should be conducted in human intervention or in vitro. Such studies have not been carried out in this work. In addition, the presence of pelargonidin was previously established by other researchers in buckwheat. Therefore, one should stick to the facts and not take priority, and it is better to read the literature on the subject.

Author Response

(The authors gave the same response as above.)

Reviewer 3 Report

I think this paper is well prepared. However, several points should be revised before processing further.

1.  Check again the references, and modify the style to fit with MDPI

2. Table 6, name of chemical should be capitalized. Some compounds like m-hydrobenzoic acid, m should be italic whilst H should be capitalized, revise this Table.

3. Write more information about the role of secondary metabolites in buckwheat and allelopathy of this plant. Buckwheat is very healthy crop and many compounds like rutin are very good for health. However your study could not find rutin, please discuss. There are some papers noted about allelopathy of buckwheat in agriculture, for instance (Allelopathy Journal 21(1): 1-11, 2003); Crop Protection 2003 22(6) 829-836; Acta Physiologiae Plantarum 2019, 41, 92; Journal of Agriculture and Food Chemistry 2007, 55 (16), 6453-6459). Information from such papers may help to increase the understanding of use of buckwheat. Please discuss about the varietal differences

4. The importance of UK and Brazil buckwheat in the world market should be provided to provide the significance of why you choose them for analyze.

5. The similarity is 33%, it should be reduced < 30%. You copied from your previously published paper 7% (Journal of Food Composition and Analysis 2021, 98, 103821), so please reduce the copied writing

Author Response

(The authors gave the same response as above.)

Round 2

Reviewer 1 Report

The manuscript has been carefully revised and are now suitable for publication.

Author Response

We would like to thank you for reviewing again the amended manuscript and for your help in improving the quality of the manuscript.

Reviewer 2 Report

Dear Authors,

The authors have significantly improved the work, but there are still a number of misconceptions that need to be filled or dispelled.

1)     In the main text and in the tables, the authors instead of providing the values ​​of the least significant difference (LSD) between the studied items, provide only the significance level p (<0.05 or <0.01) and refer to additional material that the reader does not see and will not understand all the significant differences that are discussed in the text. Such behavior of the Authors will not solve the doubts related to the discussion of significant differences between the objects. In addition, in these additional tables, the authors include only: Mean Difference, Standard Error and Significant (p value)

2)     Naming problems:

a)     buckwheat is not a proper grain but an apparent one and belongs to the class of dicots, family Polygonaceae family. Thus, both the seeds and the consumption material of buckwheat are botanically not "grains" but "nuts". The authors did not notice these botanical differences and did not correct the nomenclature.

b)     the authors confuse which elements belong to macro- and which belong to microelements. For example, they include potassium and sodium among the microelements. Please correct this.

3)       In Table 6, the mean value and standard deviation are given with the content of all secondary metabolites, but this is not sufficient to state the significance of the differences. A value for the least significant difference at least at the significance level of p <0.05 is needed. The change of the title of the work explains the authors due to the lack of a deeper statistical analysis of the obtained results. My other comments were considered the plant metabolites measured in the buckwheat samples is absent. Concentration 

Author Response

We would like to thank you for reviewing again the amended manuscript and we consider that the extra suggestion made will significantly improve the quality of the manuscript. Please see our responses in the letter attached and in amended document highlighted in red color.

Reviewer 3 Report

The authors have revised the paper extensively. However, the References have not yet well modified following the style of MDPI. In addition, all references as I noted have not yet cited, especially varietal difference of allelopathy of buckwheat. Please revised again. 

Author Response

We would like to thank you for reviewing again the amended manuscript and we consider that the extra suggestion made will improve the quality of the manuscript. Please see our responses in the letter attached and in amended document highlighted in red color.
